# Thermal dependence of the hydrated proton and optimal proton transfer in the protonated water hexamer

Félix Mouhat [1], Matteo Peria [2], Tommaso Morresi[3], Rodolphe Vuilleumier [4], Antonino Marco Saitta [2] & Michele Casula [2] ✉

Water is a key ingredient for life and plays a central role as solvent in many biochemical reactions. However, the intrinsically quantum nature of the hydrogen nucleus, revealing itself in a large variety of physical manifestations, including proton transfer, gives rise to unexpected phenomena whose description is still elusive. Here we study, by a combination of state-of-the-art quantum Monte Carlo methods and path-integral molecular dynamics, the structure and hydrogen-bond dynamics of the protonated water hexamer, the fundamental unit for the hydrated proton. We report a remarkably low thermal expansion of the hydrogen bond from zero temperature up to 300 K, owing to the presence of short-Zundel configurations, characterised by proton delocalisation and favoured by the synergy of nuclear quantum effects and thermal activation. The hydrogen bond strength progressively weakens above 300 K, when localised Eigen-like configurations become relevant. Our analysis, supported by the instanton statistics of shuttling protons, reveals that the near-room-temperature range from 250 K to 300 K is optimal for proton transfer in the protonated water hexamer.

For more than 200 years and the seminal work of von Grotthus, the properties of the hydrated proton $H^+_{(aq)}$ have intrigued the scientific community[1,2]. Despite significant advances, the exact role of the solvated proton in proton transfer (PT) reactions in chemical and biological systems is not fully elucidated yet. The textbook picture is that the hydrated proton exists as classical hydronium cation $H_3O^+$, but it looks more appropriately described as a delocalised charge defect shared by multiple molecules. The spread of this charge defect blurs the identity of the excess proton between two limiting structures, namely the Zundel[3] and the Eigen[4] ions. Indeed, the hydrated proton Infrared (IR) spectrum displays a combination of a few discrete absorption bands on top of an absorption continuum, broadly extended over the entire spectrum. Neither the symmetrically solvated hydronium $H_9O_4^+$ (Eigen) ion, nor the equally shared proton in the $H_5O_2^+$ (Zundel) ion, can individually rationalise this characteristic IR fingerprint. Models involving fast inter-conversions between these two ionic species are also shown to fail[5]. Moreover, the question of whether the hydrated excess proton and the related PT mechanism should be based on an Eigen-[6–8] or Zundel-like motif[9,10], one or the other taken as the most dominant species, has been actively debated.

To deal with these issues, Stoyanov et al.[11] have introduced the stable $H_{13}O_6^+$ species, the protonated water hexamer, which is Zundel-type in the sense that the excess proton is equally shared between two water molecules. The core of the cluster is characterised by a central oxygen–oxygen distance $d_{O_1O_2}$ that, however, is more elongated than in the Zundel cation[12]. On the other hand, recent Molecular Dynamics (MD) simulations suggest the existence of a distorted, nonsymmetric Eigen-type cation, remaining at the heart of a dynamical charge defect spanning multiple water molecules[6,13]. The protonated water hexamer represents one of the smallest protonated water clusters for which

[1]Saint Gobain Research Paris, 39, Quai Lucien Lefranc, 93300 Aubervilliers, France. [2]IMPMC, Sorbonne Université, CNRS, MNHN, UMR 7590, 4 Place Jussieu, 75252 Paris, France. [3]ECT*-Fondazione Bruno Kessler*, 286 Strada delle Tabarelle, 38123 Trento, Italy. [4]PASTEUR, Département de Chimie, École normale supérieure, PSL Research University, Sorbonne Université, CNRS, 24 Rue Lhomond, 75005 Paris, France. ✉e-mail: michele.casula@sorbonne-universite.fr

both of these characteristic binding motifs coexist[14–16]. Snapshots of the main protonated hexamer configurations are represented in Fig. 1, for both Zundel- and Eigen-like forms, while other low-lying isomers exist[17,18], but with a much lower probability to occur starting from the global minimum, due to their different topology.

The protonated hexamer Potential Energy Surface (PES) has been partially explored by IR spectroscopy[19–21] and electronic structure calculations performed within Empirical Valence Bond (EVB), Density Functional Theory (DFT), experiment-directed DFT, Møller-Plesset (MP2), and Coupled Cluster (CC) approaches, also supplemented by Machine Learning (ML) techniques[6,7,9,15,18–20,22–24], confirming that the two structures introduced above are the lowest energy isomers. The quantum nature of the proton however induces a delocalised structure on this PES.

In this work, we apply MD simulations fully retaining the nuclear quantum nature of the atoms. In particular, the quantum proton, described within the Feynman path integral (PI) approach, evolves in a very accurate PES estimated by means of Quantum Monte Carlo (QMC). This stochastic technique introduces an intrinsic noise, which affects the forces driving the ion dynamics and, consequently, the simulation temperature. Relying on the generalised fluctuation-dissipation theorem, a Langevin-based approach has been developed to address this issue for classical[25] and quantum[26] ions. It allows one to sample microscopic configurations in the canonical ensemble with a high level of quantum accuracy. Although our QMC wave function ansatz leads to the dissociation limit of two water molecules ≈ 20% (≈1 kcal/mol) off with respect to CCSD(T) values, the QMC PES is sufficiently accurate to yield structural properties and quantum-thermal distributions around equilibrium in a good agreement with CCSD(T)-derived PESs for benchmark systems such as the water dimer[27] and the Zundel ion[28]. Details of the method and of its accuracy are provided in the 'Methods' section and in Supplementary Notes I and II of the Supplementary Information (SI).

We find that the hydrogen bond (H-bond) mediated by the hydrated proton shows a remarkably low thermal expansion from zero temperature up to 300 K, with a nearly temperature-independent length that becomes shorter than the classical-ion counterpart in the [200–350] K temperature range. A non-trivial behaviour of the H-bond has also been found in H-rich crystals and ferroelectric materials, such as the potassium dihydrogen phosphate (KDP)[29], first detected by Ubbelohde in 1939 upon isotopic substitution[30]. In the latter case, the lighter the hydrogen, the shorter the H-bond. This was interpreted as a quantum manifestation of proton delocalisation, strengthening the H-bond. In the present situation, the strength of the H-bond results from a non-trivial cooperation of nuclear quantum effects (NQEs) and thermal activation, as we will show in this work. Indeed, NQEs strongly affect the

vibrational levels of the proton shuttling mode bridging the central $O_1$ and $O_2$ oxygen atoms. These levels are then thermally occupied according to the $d_{O_1 O_2}$ distance of a given configuration. We can thus distinguish three regimes (see Fig. 1): (i) "short-Zundel" configurations with the shortest $d_{O_1 O_2}$, where the proton along the shuttling mode feels a quadratic potential close enough to its energy minimum and it is perfectly shared between the two central water molecules; (ii) "elongated-Zundel" configurations for intermediate $d_{O_1 O_2}$, comprising the equilibrium distance, where a potential energy barrier starts to develop in between $O_1$ and $O_2$ and the proton is delocalised only due to NQEs; (iii) "distorted-Eigen" configurations at even larger $d_{O_1 O_2}$, where the central barrier is large enough that the hydrated proton is localised on one of the two flanking water molecules, forming an Eigen-like complex.

Here, we show that the occurrence of short-Zundel configurations is key to understand the H-bond thermal robustness and to enhance the proton transfer dynamics. Despite being energetically disfavoured by the short $d_{O_1 O_2}$ distances at the classical level, these configurations are populated thanks to the synergistic action of NQEs and temperature, yielding a sweet spot for proton transfer in the [250–300] K temperature range.

## Results
### Thermal expansion of the H-bond
To rationalise our main outcome, we first study the zero-temperature classical geometry and PES of the protonated water hexamer, and compare it with the Zundel cation. While the latter system misses a large part of water solvation effects, the former includes the full contribution of the first and second shells of the solvated proton. The zero-temperature results are reported in Supplementary Note III.1 of the SI. We simply highlight here that the Variational Monte Carlo (VMC) equilibrium $O_1$-$O_2$ distance is found to be $d_{O_1 O_2} = d_{min} = 2.3930(5)$ Å, in good agreement with MP2 calculations, the most widely used post Hartree-Fock theory to study water clusters (see Supplementary Note I.2 of the SI for a more extended comparison between VMC and MP2). VMC has a milder scale with the system size than MP2, allowing one to perform extensive calculations of the protonated hexamer. At variance with the Zundel cation[31,32], the protonated hexamer equilibrium geometry is asymmetric, implying that the global minimum is split into a double well, separated by an energy barrier between the two central water molecules. This barrier vanishes at $d_{O_1 O_2} = d_{symm} \simeq 2.38$ Å, a distance that separates the short Zundel below from the elongated-Zundel configurations above. The height of the barrier is less than 100 K (in $k_B$ units) at $d_{min}$, rapidly increasing as a function of $d_{O_1 O_2}$. We therefore expect several consequences on the hydrated proton distribution and on its mobility at finite temperature, once the NQEs are taken into account.

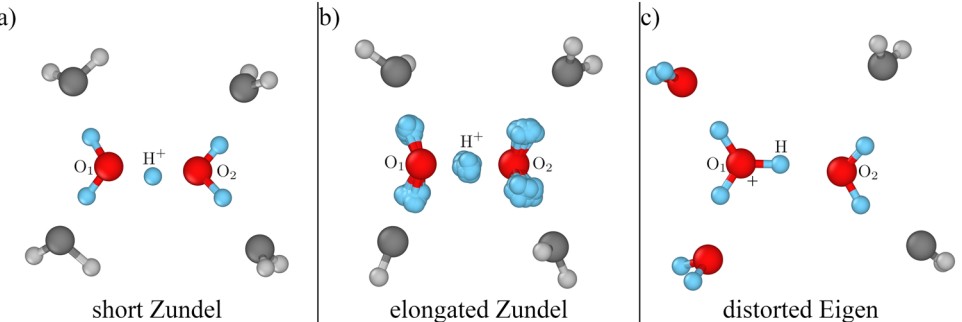

**Fig. 1 | Different regimes of the protonated water hexamer $H_{13}O_6^+$. a** Short-Zundel configuration with a Zundel center ($H_5O_2^+$) in colours and its first solvation shell (4 $H_2O$) in grey shades. **b** elongated Zundel with the quantum nature of hydrogen atoms highlighted by the full representation of its imaginary-time positions in a PI configuration. **c** distorted-Eigen configuration with an Eigen cation ($H_9O_4^+$) in colours accompanied by two solvating water molecules (2 $H_2O$) in grey shades. The $O_1$, $O_2$ and $H^+$ labels are used throughout the paper to refer to the corresponding atoms, as indicated here.

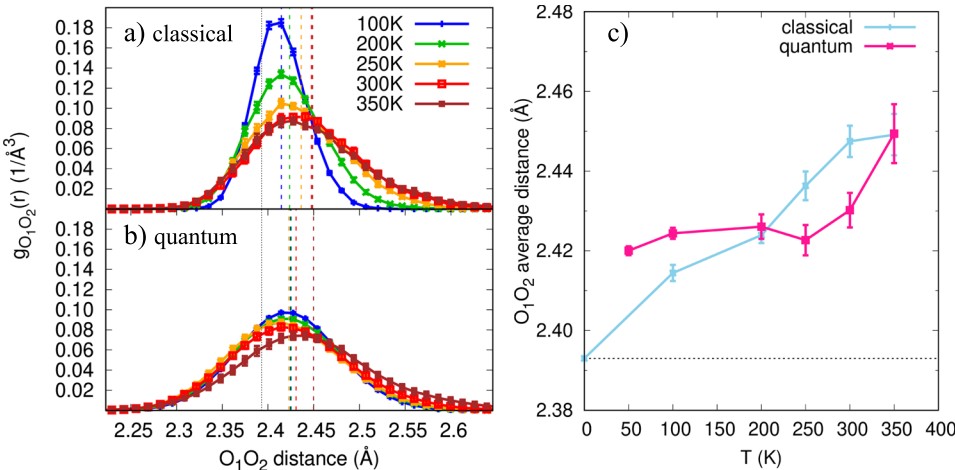

**Fig. 2 | Classical and quantum oxygen–oxygen $g_{O_1O_2}$ pair correlation functions as a function of temperature. a, b** The dashed vertical lines indicate the average $\langle d_{O_1O_2} \rangle$ distance for each simulation, at the corresponding temperature. The dotted vertical line is located at the classical equilibrium geometry. Panel (**c**) shows

the T-dependence of the $\langle d_{O_1O_2} \rangle$ average distance. The classical equilibrium geometry is represented by a short-dashed horizontal black line. At 250 K and 300 K the oxygen–oxygen distance is shortened by NQEs with respect to the classical counterpart. Source data are provided as a Source data file.

To understand how the dynamics of the hydrated proton evolves with temperature, QMC-driven ab initio MD simulations are relevant. Such calculations are carried out for both classical and quantum nuclei of the $H_{13}O_6^+$ ion, within the temperature interval T ∈ [50−350] K, thanks to the methodological developments detailed in ref. [26] and in the 'Methods' section. At these conditions, the clusters are stable during the simulated time frame (≈30 ps), allowing us to access the thermal properties of the hydrated proton and the $O_1H^+O_2$ bond over an extended temperature range. This is the advantage of performing accurate computer simulations for this system, particularly if an experimental investigation of the cluster will require a more extended lifetime, with the risk of molecular evaporation at the highest temperatures studied here. The details of these challenging calculations, such as the number of iterations and their computational costs at a given temperature, are reported in Supplementary Table 4 of the SI.

From our QMC-MD simulations, we extract the normalised Pair Correlation Functions (PCFs) $g_{O_1O_2}$ for the two oxygen atoms $O_1$ and $O_2$ of the cluster core (Fig. 2). The expected broadening of the PCFs due to nuclear quantisation is significant over the whole temperature range (Fig. 2b). Only at temperatures as high as 350 K, the classical $g_{O_1O_2}$ (Fig. 2a) starts resembling the quantum distribution. This implies that the NQEs cannot be neglected for temperatures up to this value, above ambient conditions. We also notice that, when comparing to the Zundel ion results[33], the peak position is shifted up by at least ~0.01 Å. Thus, it appears that the $H_{13}O_6^+$ cluster frequently adopts elongated-Zundel configurations[8,34,35] at the lowest temperatures considered here. This is at variance with the protonated water dimer, where the hydrated proton lives in a single minimum symmetrically located between the two water molecules.

Focusing our attention to $\langle d_{O_1O_2} \rangle$ (Fig. 2c), its classical and quantum behaviours are remarkably different as a function of temperature. On the one hand, the classical $d_{O_1O_2}$ keeps increasing with temperature, as more energy is given to the intermolecular vibration modes. On the other hand, the quantum $d_{O_1O_2}$ displays a nearly flat behaviour with the cluster temperature, up to 300 K. This very low thermal expansion extended over a wide temperature range leads to a temperature regime where $d_{O_1O_2}$ for the quantum system become shorter than the classical values at the same temperatures. This is clearly seen in Fig. 2c. We will come back to this point later.

Finally, as the temperature further increases, the NQEs reduction weakens the central H-bond strength. Consequently, $d_{O_1O_2}$ spreads out, due to stochastic fluctuations of the core and the solvent, and a

more classical regime is reached, when the averaged $d_{O_1O_2}$ values for classical and quantum nuclei meet again. The PCF distributions display longer tails, with more configurations covering regions with $d_{O_1O_2} \in [2.5 − 2.7]$ Å, and the peak position rapidly shifts to larger values. Configurations with such a large $\langle d_{O_1O_2} \rangle$ are of distorted-Eigen type[8,36].

### A cooperative thermal-quantum species: the short-Zundel ion

To refine our structural analysis, we compute the bidimensional distribution function $\rho_{2D}$, which correlates the oxygen–oxygen ($O_1O_2$) and the oxygen–proton ($O_{1/2}H^+$) distances, and study its temperature dependence $\rho_{2D} = \rho_{2D}(T)$. In Fig. 3, we show the contour plot of the temperature-driven $\rho_{2D}$ variation (see also Supplementary Note III.2 of the SI). By taking $\rho_{2D}(250 \text{ K})$ as reference, four temperature variations are explored: 100 K, 200 K, room temperature (RT), and 350 K (from the top to the bottom of Fig. 3).

In the classical protonated hexamer (Fig. 3, left column), rising the temperature from 250 K up to 350 K tends to stretch $\langle d_{O_1O_2} \rangle$, by promoting configurations from the elongated Zundel (blue central distribution with $d_{O_1O_2} \in [2.38,2.5]$ Å in Fig. 3) to an Eigen-like arrangement with larger $d_{O_1O_2}$ and a proton much more localised on one of the two central oxygen atoms (red wings). The situation is reversed at lower temperatures (100 K and 200 K) if compared to the 250 K reference, with positive (red) variations in the elongated Zundel and negative (blue) variations in the wings. Thus, for classical nuclei, there is a progressive depletion of the elongated Zundel and a corresponding population of the distorted-Eigen wings upon temperature rise. Short-Zundel configurations, highlighted in Fig. 3 by a grey background, seem to play a very marginal role in the temperature-driven density distribution shift.

The scenario is strikingly different with quantum nuclei (right column), particularly at the lowest temperatures (100 K and 200 K). In this regime, distorted-Eigen configurations are barely populated or depleted, and the density shift upon rising temperature takes place between the elongated-Zundel region and the short-Zundel sector. The latter is significantly more populated at 250 K than at lower temperatures at the expense of the elongated Zundel, which instead loses density with respect to the classical counterpart at the same temperature.

In the higher-temperature limit, at 350 K, NQEs are less relevant and, by consequence, the classical and quantum variations have a qualitatively similar behaviour. In both classical and quantum case, we notice the presence of red wings at large oxygen–oxygen distances

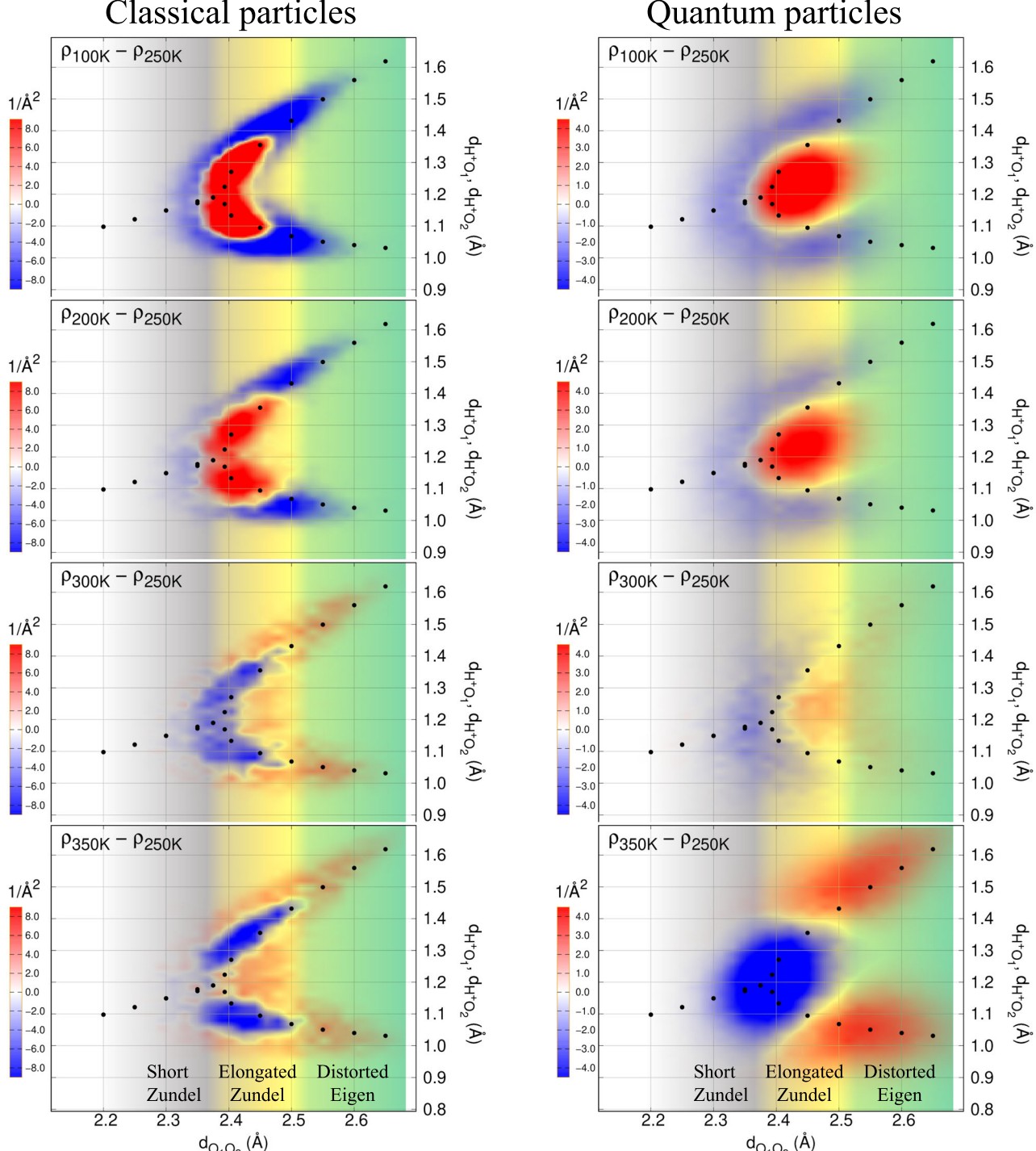

**Fig. 3 | Bidimensional oxygen–oxygen/oxygen–proton distributions.** Difference between bidimensional oxygen–oxygen/oxygen–proton distributions $\rho_{2D}$ obtained by QMC-driven LD simulations for classical (left panels) and quantum (right panels) particles, computed at different temperatures. The bidimensional distribution computed at 250 K is taken as reference. Positive (negative) regions are in red (blue) colour. The black filled circles correspond to the zero-temperature equilibrium geometries of the $H_{13}O_6^+$ ion at a fixed $d_{O_1O_2}$ distance. The coloured background highlights the three different regimes explained in the paper: the short Zundel (grey), the elongated Zundel (yellow), and the distorted Eigen (green) species. Source data are provided as a Source Data file.

($d_{O_1O_2} \in [2.5, 2.7]$ Å), which are the signature of thermally activated Eigen-like states, with a strongly localised proton. This is related to less frequent elongated-Zundel configurations, indicated by the depleted distribution for $d_{O_1O_2} < 2.5$ Å, confirming that the distorted-Eigen configurations are indeed promoted by high temperature. For quantum nuclei, the corresponding depletion goes well below the elongated-Zundel region, by touching also short-Zundel configurations, down to

$d_{O_1O_2} \sim 2.3$ Å, at variance with the classical case, where the short-Zundel configurations are not involved.

To interpret these results, we first construct an accurate effective potential by projecting the full PES, computed during QMC-driven classical MD calculations, onto the degrees of freedom mostly relevant to understand the dynamics of the hydrated proton. These are the $d_{O_1O_2}$ distance and the proton sharing coordinate $\delta$, referenced to the

midpoint of the $O_1H^+O_2$ complex: $\delta \equiv \tilde{d}_{O_{1/2}H^+} - d_{O_1O_2}/2$, with $\tilde{d}_{O_{1/2}H^+}$ the $O_{1/2}$-$H^+$ distance projected onto the $O_1O_2$ direction. The resulting two-dimensional (2D) potential is $V_{2D} = V_{2D}(d_{O_1O_2},\delta)$. We refer the reader to Supplementary Notes IV and V of the SI for technical details about the PES projection. We highlight that the potential $V_{2D}$ is derived here at VMC quality. We also notice that $\delta$ is the vibrational coordinate of the proton shuttling mode, while $d_{O_1O_2}$ is related to the stretching mode of the two water molecules in the cluster core.

Given $V_{2D}(d_{O_1O_2},\delta)$, we then proceed to quantize the variable $\delta$. Indeed, while $d_{O_1O_2}$ can be taken as classical, for it is related to the motion of heavier oxygen atoms of mass $m_O$, the $\delta$ coordinate must be quantised, owing to the light mass ($m_H$) of the hydrated proton. At the leading order in $2m_H/(m_O + m_H)$, we separate the stretching mode from the shuttling one, by invoking an adiabatic Born-Oppenheimer type of approximation for the two species[37]. We finally solve quantum-mechanically the Hamiltonian of a proton in the potential $V_\delta \equiv V_{2D}(\alpha,\delta)|_{\alpha=d_{O_1O_2}}$ at fixed $d_{O_1O_2}$ value. In Fig. 4a–c we plot the ground state distribution and eigenvalues obtained for three distances, i.e. at $d_{O_1O_2} = 2.375$ Å, in the short-Zundel region close to the boundary between the short and the elongated Zundel, at $d_{O_1O_2} = 2.495$ Å, in the elongated-Zundel region close to the frontier between the elongated Zundel and the distorted Eigen, and finally at $d_{O_1O_2} = 2.585$ Å, deep into the distorted Eigen regime.

One can notice three different quantum behaviours of the vibrational shuttling mode, that provide a more quantitative ground to the three-regime distinction made at the beginning. In the short Zundel, $V_\delta$ is indeed a quadratic potential with a single minimum at the core center, which widens as $d_{O_1O_2}$ gets close to $d_{symm} \simeq 2.38$ Å, a distance where it becomes quartic because its curvature falls to zero before changing sign. The ground state energy, i.e. the zero point energy (ZPE) of the shuttling mode, decreases as the potential widens, as reported in Fig. 4d. In the elongated Zundel, a central barrier starts to develop, with a ground-state proton distribution that stays uni-modal thanks to a ZPE larger than its height, till $d_{O_1O_2} \simeq 2.5$ Å, where the ZPE equals the barrier height. In this regime, for $d_{O_1O_2} \in [d_{symm}, 2.5$ Å$]$, the ZPE is particularly small, due to the quartic nature of $V_\delta$, and weakly $d_{O_1O_2}$-dependent, as shown in Fig. 4d. Finally, for $d_{O_1O_2} > 2.5$ Å, we enter the distorted-Eigen regime, with an even larger central barrier (>1000 K), such that the quantum proton is instantaneously localised in one of the two wells, and its distribution is then bimodal. The ZPE starts to rise again as $d_{O_1O_2}$ is stretched, with a slope steeper—in absolute value— than the ZPE decrease in the short Zundel, because it is now set by the much deeper lateral minima of the double-well potential. This can be seen again in Fig. 4d.

We can now correct the classical $O_1$-$O_2$ potential, defined as $V_{O_1O_2} \equiv V_{2D}(d_{O_1O_2},\delta)|_{\delta=\delta_{min}}$, where $\delta_{min}$ is the $V_{2D}$ minimum at fixed $d_{O_1O_2}$ value, by adding the ZPE $\forall d_{O_1O_2}$, obtained from the quantisation of the shuttling mode $\delta$. The resulting potential is plotted in Fig. 4e. Remarkably, the anharmonic classical $V_{O_1O_2}$ potential becomes harmonic after ZPE-correction. It is a consequence of the much larger ZPE in the distorted-Eigen configurations than in the short Zundel, which compensates for the underlying $V_{O_1O_2}$ anharmonicity. This rationalises two main features. On the one hand, it explains the very low thermal expansion of $\langle d_{O_1O_2} \rangle$, being the average position in a harmonic potential temperature-independent. On the other hand, it proves that NQEs enhance the occurrence of short-Zundel configurations upon heating, while the distorted Eigen is penalised by its large ZPE with respect to the classical counterpart. The enhancement of the

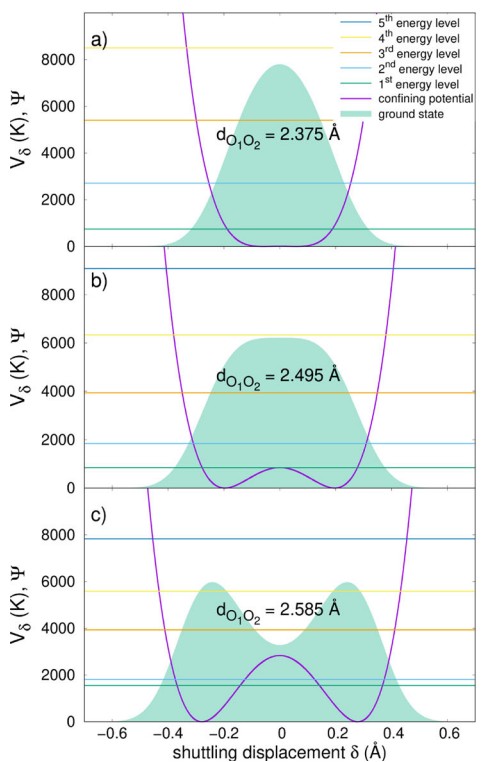

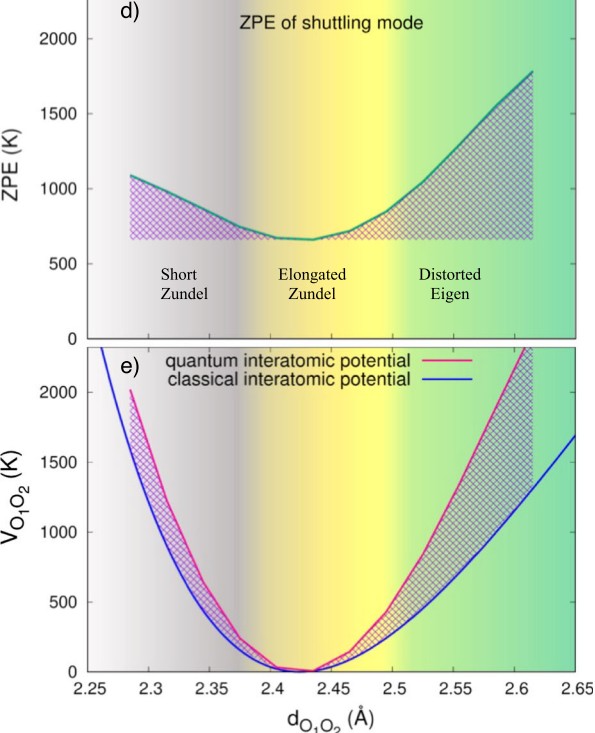

**Fig. 4 | NQEs on the shuttling mode, and their impact on the $O_1$-$O_2$ interatomic potential $V_{O_1O_2}$.** We quantize the proton shuttling mode $\delta$, defined as the displacement along the segment connecting the two oxygen atoms in the core of the cluster from its midpoint position. We study the ground-state wave function and the first 5 eigenvalues for the confining potential $V_\delta$, as a function of $d_{O_1O_2}$. Panels (**a**), (**b**) and (**c**) report the ground state wave function and the lowest 5 energy levels for $d_{O_1O_2} = 2.375, 2.495$ and $2.585$ Å, respectively. In panel (**d**), the variation of the zero-point (ground-state) energy (ZPE) as a function of $d_{O_1O_2}$ is explicitly plotted. While the ZPE dependence is very flat in the elongated-Zundel region (depicted by the yellow shaded area), the ZPE increases in both short-Zundel (grey shaded area) and distorted-Eigen (green shaded area) regions, with a much steeper slope in the latter. In panel (**e**), the ZPE is added to the classical interatomic potential $V_{O_1O_2}$ (solid blue line) to yield the quantum-corrected effective interatomic potential (solid dark-pink line) between the two inner oxygen atoms.

occurrence of Zundel configurations by NQEs is also revealed by the population analysis presented in Supplementary Note VI and Supplementary Fig. 17 of SI. This is also in agreement with a similar analysis carried out in ref. 6 in bulk water.

Above RT, the distorted Eigen configurations will eventually become dominant again. This can be understood within this framework as well. Indeed, thermal excitations are energetically more available in the distorted Eigen, where the spacing between the ZPE and the first-excited state shrinks, and higher excited states are piled up more densely than in the short and elongated Zundel (see Fig. 4a–c). The full thermal dependence of the short Zundel, elongated Zundel and distorted Eigen populations is reported in Supplementary Fig. 16 of Supplementary Note VI in SI.

### Optimal proton transfer from instantons statistics

The analysis made so far highlights the paramount importance of the NQEs to set the non-trivial temperature behaviour of the $H_{13}O_6^+$ cluster. At this stage, direct information about the excess proton dynamics along the QMC-PIMD trajectory is necessary to estimate more quantitatively its impact on the PT processes occurring in the system.

One way to achieve this goal is by analysing the statistics of selected transition-state (TS) configurations, defined by means of instanton theory. Within the PI formalism, the instanton path seamlessly connects the reactants and products minima, along the minimal action trajectory, periodic in the quantum imaginary time $\tau = \beta\hbar$[38]. It provides a generalisation of the TS theory for anharmonic quantum systems[39], and it has been very recently applied in a QMC framework[40,41], by efficiently recovering the proper scaling of ground-state tunnelling rates. TS configurations are therefore identified as those where each half of the instanton path is located on either side of the central $O_1O_2$ midpoint, sampled during the QMC-PIMD dynamics.

With the aim at resolving the contribution of the three different regimes to the PT dynamics, we collect the instanton events and compute their statistical distribution as a function of $d_{O_1O_2}$. We plot the instanton density distribution function in Fig. 5a at various temperatures. To deepen our analysis, we compute also the cumulative density distribution function in Fig. 5b, after normalising it based on the algorithmic frequency of the instanton occurrences, as counted during our QMC-PIMD simulations. Although this does not give direct access to real-time quantities, the ring-polymer MD with Langevin

thermostat has been shown to yield physically reliable information on frequencies and frequency variations[42]. Note that the coupling with the Langevin thermostat is kept constant across the full temperature range analysed here[42]. The fully integrated frequency distribution gives the total proton hopping frequency, plotted in Fig. 5c as a function of temperature. This shows a clear maximum located in the [250−300] K temperature range. Consequently, we expect the hydrated proton mobility to be optimal in a near-RT window, with a maximised Grotthus diffusion. To understand the source of this temperature sweet spot, in the same panel (c) we plot the contribution to the total frequency of instanton events occurring in the short-Zundel region. This is yielded by the cumulative frequency distribution of panel (b) evaluated at the boundaries between short and elongated Zundel, i.e. at $d_{O_1O_2} = d_{symm}$. The short-Zundel contribution to the total frequency shows a peak of the same intensity as the total one in the same temperature range, clearly pointing to the key role played by thermally activated short-Zundel configurations to the PT dynamics. The short-Zundel arrangement enables instantaneous proton jumps between the two sides of the cation, since there is no barrier to cross. Thus, the sweet spot constitutes the best compromise between acquiring enough thermal energy to access short-distance configurations, boosted by NQEs, and controlling the amplitude of the chemical (covalent or H-) bonds fluctuations, that might trap the proton into an asymmetric well. Indeed, at larger temperatures (>300 K), the onset of distorted-Eigen and the corresponding fall of short-Zundel configurations localize the hydrated proton around its closest oxygen atom, thus reducing its shuttling probability. A similar non-monotonous PT behaviour has experimentally been found in bulk water by assessing the limiting conductivities of the $H_3O^+$ and $D_3O^+$ species[43]. Thanks to these measures, performed at 20 MPa, the excess molar conductivities due to PT have been estimated. They show a peak located at a temperature in between 420 K and 430 K. In this temperature range and at the pressure conditions of the experiment, the water density is only 7–8% smaller than the standard conditions[44], a regime comparable to the one of our cluster.

Beside this PT mechanism, which is adiabatic in nature and driven by the synergy of ZPE and thermal effects, NQEs could also contribute to the proton diffusion by means of instantaneous tunnelling, which can further accelerate the PT dynamics. By computing the root-mean-square (RMS) displacement correlation

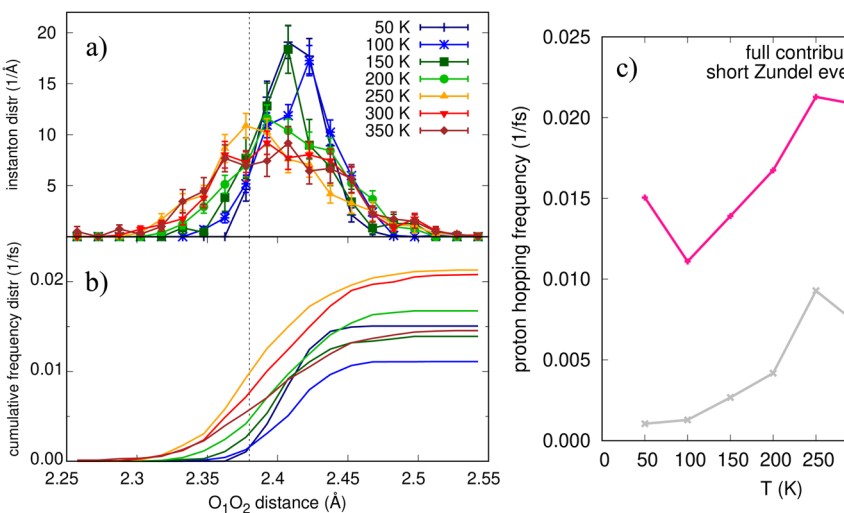

**Fig. 5 | Instanton statistics and proton hopping frequency. a** Instanton distribution resolved as a function of the $d_{O_1O_2}$ distance for different temperatures. **b** Cumulative distribution of (**a**) normalised by the occurrence frequency of the instanton (proton hopping) events during the PIMD simulations. **c** Proton hopping frequency as a function of temperature, together with the contribution coming

from the short Zundel configurations, with $d_{O_1O_2} < d_{symm} = 2.38$ Å. The $d_{symm}$ value is reported as vertical dashed line in (**a**) and (**b**). Here, we report simulations performed also at 400 K, a temperature at which the cluster is still stable or metastable. Source data are provided as a Source data file.

functions[45] over the instantons population, we verified that tunnelling events could take place only in the distorted Eigen and in the intermediate temperature range (see Supplementary Note VII of SI for a detailed analysis). This additional PT channel has however a marginal effect with respect to the main mechanism unveiled here. Indeed, Fig. 5c shows that the sweet spot is mainly due to PT events originating in short-Zundel configurations, where quantum tunnelling is not relevant.

## Discussion

Using highly accurate QMC-PIMD simulations of the $H_{13}O_6^+$ cation at finite temperature, we found a remarkably low thermal expansion of the protonated water hexamer core. It stems from a cooperative action of both NQEs and thermal effects, which leads to the emergent behaviour of short-Zundel species as PT booster, where the excess proton is perfectly shared between two neighbouring water molecules. The relevance of short-Zundel configurations is enhanced by NQEs, which instead penalize the distorted-Eigen states, having a larger ZPE. In the intermediate temperature range, comprising RT, the occurrence of short-Zundel events is maximised by thermal population, leading to a sweet spot in the PT dynamics. Around these temperatures, distorted-Eigen states can still contribute to PT with quantum tunnelling processes, although occurring at much lower rates. The cluster core spreads out again at larger temperatures, as soon as stronger thermal fluctuations favour the formation of more classical distorted-Eigen structures, where the proton gets strongly localised in one of the flanking molecules.

The short-Zundel quantum species is crucial for an efficient proton diffusion, as the shortness of its structure enables a fast charge redistribution during the adiabatic PT process. Recent progress in ultrafast broadband two-dimensional (2D) IR spectroscopy[46,47] allowed to probe the vibrational properties of protonated water at vibrational frequencies around the hydrated proton stretching mode, by measuring the lowest-lying excitations in the mid-infrared continuum[47]. These state-of-the-art experiments revealed a strongly inhomogeneous behaviour of the pump-probe spectra, implying large structural distributions in proton asymmetry and $O_1O_2$ distance. Therefore, the traditional "Zundel limit"[3] needs to be revisited and extended, in order to cover the broad range of structures detected experimentally[9,48]. In particular, the occurrence of qualitatively different short hydrogen-bond configurations, straightforwardly connected with the short-Zundel species described here, has been detected and highlighted in a recent fully solvated $(HF_2)^-(H_2O)_6$ experiment through femtosecond 2D IR spectroscopy in ref. 49. The present work crucially extends those findings by providing a temperature resolved analysis of the short hydrogen-bond events and by revealing their fundamental relation with the PT dynamics.

While proton transfer and proton transport occur in a variety of environments, from solutions to membrane proteins and fuel-cell membranes, the protonated water hexamer is one of the smallest clusters to incorporate most of the PT experimental features and solvation effects at the leading order. According to ref. 50, one more hydration layer is needed to reach the water bulk limit. From this viewpoint, the hexamer is close to that limit, and some relevant effects, emerging already at this size, can be transferred to larger systems. Our findings thus call for further efforts to explore the temperature behaviour of the proton dynamics and transport both in aqueous systems and in other extended environments, by keeping the same accuracy as the one delivered by our QMC-driven PIMD approach in the protonated water hexamer. That goal could be achieved by training efficient atomistic ML potentials on QMC[51,52] or other high-quality datasets[24,53], which could allow one to generalise quantitatively the PT behaviour unveiled here to a wider class of aqueous systems.

## Methods

### Zero-temperature electronic structure calculations

Before running finite-temperature calculations, we build a quantum Monte Carlo (QMC) variational wave function. The molecular dynamics (MD) will develop on the potential energy surface (PES) generated by the variational energy of this wave function. All zero- and finite-temperature calculations have been carried out with the TurboRVB code[54]. We choose the variational ansatz such that the chemical accuracy (1 kcal/mol) in the binding energy of the Zundel ion and water dimer is reached. We highlight the fact that benchmarking the binding energy is much stricter than taking energy differences of geometries around the minimum, because the configurations involved in the binding energy are very different.

The variational wave function $|\Psi_q\rangle$ we used in our work is written as a Jastrow Antisymmetrised Geminal Power (JAGP) product[55]

$$\Psi_q(\mathbf{x}_1,\ldots,\mathbf{x}_{N_{el}}) = J_q(\mathbf{r}_1,\ldots,\mathbf{r}_{N_{el}})\Psi_{AGP,q}(\mathbf{x}_1,\ldots,\mathbf{x}_{N_{el}}). \quad (1)$$

The set $\{\mathbf{x}_i = (\mathbf{r}_i, \sigma_i)\}_{i=1,\ldots,N_{el}}$ represents spatial and spin coordinates of the $N_{el}$ electrons, and $\mathbf{q}$ is the vector of nuclear coordinates.

We report here the main ingredients of the JAGP wave function[56–59]. The Bosonic Jastrow factor is written as a product of one-body, two-body and three/four-body terms $J_q = J_{1,q}J_{2,q}J_{3,q}$. The one-body term reads

$$J_{1,q} = \exp\left(-\sum_i^{N_{el}}\sum_j^N \left(2Z_j\right)^{3/4} u\left(\left(2Z_j\right)^{1/4}\left|\mathbf{r}_i - \mathbf{q}_j\right|\right)\right) \quad (2)$$

with $u(|\mathbf{r} - \mathbf{q}|) = \frac{1 - e^{-b|\mathbf{r}-\mathbf{q}|}}{2b}$ and $b$ is a variational parameter, which satisfies the electron-ion Kato cusp conditions. $N$ the number of atoms and $Z_j$ the electric charge of the $j$-th atom. In the protonated hexamer Hamiltonian, the hydrogen keeps the bare Coulomb potential, while the oxygen atoms are replaced by the Burkatzki-Filippi-Dolg (BFD) pseudopotential[60], smooth at the electron-ion coalescence points. Thus, $J_{1,q}$ is applied only to the hydrogen atom. The two-body correlations are dealt with by the higher-order Jastrow factors. The two-body Jastrow factor is defined as

$$J_{2,q} = \exp\left(\sum_{i<j}^{N_{el}} u\left(\left|\mathbf{r}_i - \mathbf{r}_j\right|\right)\right), \quad (3)$$

where $u$ is a function of the same form as in Eq. (2), but with a different variational parameter. The three-four body Jastrow factor is

$$J_{3,q} = \exp\left(\sum_{i<j}^{N_{el}} \Phi_{J_q}\left(\mathbf{r}_i,\mathbf{r}_j\right)\right), \quad (4)$$

with

$$\Phi_{J_q}(\mathbf{r}_i,\mathbf{r}_j) = \sum_{a,b}^N \sum_{\mu,\nu}^{N_{basis}^J} g_{\mu,\nu}^{a,b} \Psi_{a,\mu}^J(\mathbf{r}_i - \mathbf{q}_a)\Psi_{b,\nu}^J(\mathbf{r}_j - \mathbf{q}_b), \quad (5)$$

where $N_{basis}^J$ is the number of the basis set functions $\Psi_{a,\mu}^J$. We used optimally contracted geminal embedded orbitals (GEOs)[61] as basis set, expanded over a primitive O(3s,2p,1d) H(2s,1p) Gaussian basis. The convergence study of the water dimer binding energy as a function of the Jastrow GEO expansion is reported in Supplementary Note I.1 of the SI.

The Fermionic part of the wave function is expressed as an antisymmetrised product of the spin singlet geminals or pairing (AGP) functions $\Phi_{\mathbf{q}}(\mathbf{x}_i, \mathbf{x}_j)$:

$$\Psi_{AGP,\mathbf{q}}(\mathbf{x}_1, \ldots, \mathbf{x}_{N_{el}}) = \hat{A}\left[\Phi_{\mathbf{q}}(\mathbf{x}_1, \mathbf{x}_2), \ldots, \Phi_{\mathbf{q}}\left(\mathbf{x}_{N_{el}-1}, \mathbf{x}_{N_{el}}\right)\right]. \quad (6)$$

The spatial part $\phi_{\mathbf{q}}(\mathbf{r}_i, \mathbf{r}_j)$ of the spin singlets $\Phi_{\mathbf{q}}(\mathbf{x}_i, \mathbf{x}_j)$ is expanded over $N_{basis}^{AGP}$ optimally contracted GEOs, such that

$$\phi_{\mathbf{q}}(\mathbf{r}_i, \mathbf{r}_j) = \sum_{a,b}^{N} \sum_{\mu,\nu}^{N_{basis}^{AGP}} \lambda_{\mu,\nu}^{a,b} \bar{\Psi}_{a,\mu}^{AGP}(\mathbf{r}_i - \mathbf{q}_a) \bar{\Psi}_{b,\nu}^{AGP}\left(\mathbf{r}_j - \mathbf{q}_b\right). \quad (7)$$

The AGP GEOs are linear combination of primitive O(5s5p2d) H(4s2p) Gaussian basis functions. We highlight that the Fermionic part is fully optimised at the QMC level for each MD step, and not generated by on-the-fly DFT calculations.

The GEOs are very effective in reducing the total number of variational parameters, by keeping a high level of accuracy. This makes the wave function optimisation[55,62,63] much more efficient. Dealing with a compact wave function is very important, if one wants to use it as variational ansatz in a MD simulation, because in a MD framework the wave function needs to be optimised at every MD iteration. Indeed, the wave function optimisation is by far the most time-consuming step of our QMC-driven MD approach.

Previous works on the Zundel ion[26,59] found that the optimal balance between accuracy and computational cost for the determinantal part is reached by the O[8]H[2] contracted GEO basis, in self-explaining notations. As the protonated water hexamer is a very similar system, in this work we used the same O[8]H[2] GEO contraction for the AGP part. Moreover, we further simplified the variational wave function previously developed for the Zundel ion, by contracting also the Jastrow basis set, using the same GEO embedding scheme. We found that the O[6]H[2] GEO basis set for the Jastrow factor is a very good compromise between accuracy and number of parameters, as we checked in the water dimer (see Supplementary Note I.1 of the SI). The final accuracy is about 1 kcal/mol in the dissociation energy of the water dimer, and supposedly it is much higher around the stable geometries of water clusters. Thus, we used the O[6]H[2] GEO basis set for the Jastrow factor, and the O[8]H[2] GEO basis for the AGP part in all our subsequent MD simulations. For the protonated water hexamer, this results into a total number of 6418 variational parameters, comprising $g_{\mu,\nu}^{a,b}$ in Eq. (5), $\lambda_{\mu,\nu}^{a,b}$ in Eq. (7), the parameters of the homogeneous one-body (Eq. (2)) and two-body (Eq. (3)) Jastrow factors, and the linear coefficients of the Jastrow and determinantal basis sets.

## Finite-temperature calculations

**Path integral Langevin dynamics.** At finite temperature, we carried out path integral Langevin dynamics simulations to include NQEs. To do so, we used the recently developed algorithm published in ref. 26, which combines a path integral approach with very accurate Born-Oppenheimer (BO) forces computed by QMC. It is an efficient approach, alternative to the coupled electron ion Monte Carlo (CEIMC) developed by Ceperley and coworkers[64–66]. The intrinsic noise of the QMC force estimator is treated by the noise correction scheme developed in refs. 25,26,67, which is based on the fulfilment of the fluctuation-dissipation theorem. This implies that the friction matrix $\gamma$ governing the dumped dynamics is related to the random force $\boldsymbol{\eta}$ via the α matrix:

$$\alpha(\mathbf{q}) = 2k_B T \gamma(\mathbf{q}), \quad (8)$$

$$\langle \eta_i(t)\eta_j(t')\rangle = \alpha_{i,j}(\mathbf{q})\,\delta(t-t'), \quad (9)$$

with $\mathbf{q}$ the vector of nuclear coordinates. The $\mathbf{q}$-dependence comes from the QMC noise correction, implemented through the relations:

$$\alpha_{i,j}(\mathbf{q}) = \gamma_{BO}/(2k_B T)\delta_{i,j} + \Delta_0 \alpha_{i,j}^{QMC}(\mathbf{q}) \quad (10)$$

$$\alpha_{i,j}^{QMC}(\mathbf{q}) = \langle(\mathbf{f}_i(\mathbf{q}) - \langle\mathbf{f}_i(\mathbf{q})\rangle)\rangle\langle(\mathbf{f}_j(\mathbf{q}) - \langle\mathbf{f}_j(\mathbf{q})\rangle)\rangle, \quad (11)$$

where $\alpha^{QMC}$ in Eq. (11) is the covariance matrix of QMC ionic forces, measuring their stochastic fluctuations, and $\gamma_{BO}$ and $\Delta_0$ parameters taking values for an optimal Langevin dynamics. The random force $\boldsymbol{\eta}$ is then used to thermalise the system to a target temperature, according to the Langevin thermostat of Eq. (8).

The quantum particles are described by necklaces extended in the imaginary time interval $[0, \hbar\beta]$, with $\beta = 1/(k_B T)$, following the quantum-to-classical isomorphism. This imaginary time interval is divided into $M$ slices, the so-called "beads", leading to an effective classical system of $NM$ particles. The path integral Langevin dynamics we developed in ref. 26 is very efficient, because the quantum harmonic forces and the Langevin thermostat−thermalising the quantum degrees of freedom− are evolved together by means of an exact propagator, without Trotter breakup. The evolution of quantum particles in a thermal bath à la Langevin represents a quantum Ornstein-Uhlenbeck dynamics. Therefore, we dubbed our integration scheme as "path integral Ornstein Uhlenbeck dynamics (PIOUD)". The algorithm is detailed in ref. 26.

In order to evolve the system and to sample the thermal quantum partition function, nuclear forces must be estimated at each iteration by computing the gradients $\mathbf{f} = -\boldsymbol{\nabla}_{\mathbf{q}}V(\mathbf{q})$. The potential energy surface $V(\mathbf{q})$ is evaluated by VMC, namely:

$$V(\mathbf{q}) = \frac{\langle\Psi_{\mathbf{q}}|H(\mathbf{q})|\Psi_{\mathbf{q}}\rangle}{\langle\Psi_{\mathbf{q}}|\Psi_{\mathbf{q}}\rangle}, \quad (12)$$

where $|\Psi_{\mathbf{q}}\rangle$ is the QMC wave function, which minimizes the expectation value of $H(\mathbf{q})$ for each bead configuration $\mathbf{q}^{(k)}$, according to the Born-Oppenheimer approximation.

The electronic variational wave function depends on the coordinates $\mathbf{q}^{(k)}$ of the $k$-th bead in two ways. Directly, through the explicit dependence on the ion positions provided by the localised basis set, and in an indirect way, through the wave function parameters optimised at each $\mathbf{q}^{(k)}$, in compliance with the Born-Oppenheimer approximation. While the former dependence is of leading order, the latter can be neglected in a first approximation. A clever way to do so is to average the optimal parameters across different beads, by gaining a significant fraction of statistics in the QMC energy minimisation. More precisely, the beads are gathered in groups of $N_{groups}$ members each, to share the same set of wave function parameters. This is called "bead grouping approximation", and it has been introduced in ref. 26.

Once the number of groups $N_{groups}$ is set for the beads, the electronic wave function is optimised at each new ionic position generated by the dynamics. This is done by energy minimisation via the most advanced optimisation techniques[57,63]. Between two consecutive steps of ion dynamics, one needs to perform $N_{opt}$ steps of energy minimisation, in an iterative fashion. $N_{opt}$ must be large enough to converge the wave function for each new ionic configuration. Thus, this parameter is tuned such that the BO approximation is fulfilled, and the dynamics follows the correct PES along the PIOUD trajectories.

During the dynamics, the GTO exponents in both the Jastrow and the AGP part of the wave function are kept frozen to make the simulation stable. Due to the continuity of the nuclear trajectories, the number of energy minimisation steps is significantly smaller than the one required for a wave function optimisation from scratch.

There is a set of sensitive parameters one needs to tune to have stable and unbiased simulations. They are:

(i) Convergence for quantum effects set by M. In our calculations, we used M = 32 for T = 250–400 K, M = 64 for T = 150–200 K, and M = 128 for T = 50–100 K. The bead grouping approximation is made with $N_{group} = 1$. Therefore the whole ring shares the same wave function parameters, except for the nuclear coordinates;

(ii) Time step $\delta t$ for the integration of the equations of motion. We used $\delta t = 1$ fs for a controlled time integration error, yielding a difference between the virial and primitive estimators of the quantum kinetic energy below a 25 mHa threshold along all the trajectories;

(iii) A stable target temperature is reached despite the QMC noise by setting the parameters $\gamma_{BO}$ and $\Delta_0$, defining the $\alpha$ matrix in Eq. (10). We used $\gamma_{BO} = 0$ and $\Delta_0 = 0.5\delta t$, optimal values for other similar systems, such as the Zundel ion[26]. Indeed, the simulation is efficient when the damping in the BO sector is minimised. The thermalisation of the system is guaranteed thanks to the optimal damping condition[68] applied to the internal modes of the ring-polymer, with the damping parameter of the center-of-mass translational modes set to 0.231 fs$^{-1}$;

(iv) To enforce the fulfilment of the BO approximation, we allowed for $N_{opt} = 5$ iterative wave function optimisation steps at each MD iteration, such that the electronic energy minimum is reached within the statistical accuracy for every ionic configuration, and the PES is sampled without stochastic bias.

**Classical Langevin dynamics.** For classical MD calculations, we used an improved variant of the original algorithm developed by Attaccalite and Sorella[25]. This variant has been detailed in refs. [26,67]. It includes a better integration scheme, involving a Langevin noise touching both coordinates and momenta, which are therefore correlated.

As in the PIOUD calculations, relevant parameters are the time step $\delta t = 1$ fs, the QMC noise correction parameters $\gamma_{BO} = 0.2$ fs$^{-1}$ and $\Delta_0 = \delta t$, and the number of QMC optimisation steps per nuclear iteration $N_{opt} = 5$.

### Error estimates
Datapoints obtained from stochastic sampling, based either on the electron or on the joint electron-nuclear distribution, are provided with the associated stochastic error bars, which are obtained via the reblocking technique[69]. The error bars on the fitting parameters plotted in Supplementary Fig. 14 of the SI are yielded by the error propagation in the Levenberg-Marquardt solution of the nonlinear least squares curve-fitting problem. In all cases, they represent the standard deviation of the data.

### Reporting summary
Further information on research design is available in the Nature Portfolio Reporting Summary linked to this article.

## Data availability
Source data are provided with this paper.

## Code availability
The TurboRVB code[54] used to carry out our QMC-driven PIMD calculations is available under the GPLv3 license. It can be downloaded from the following open source repository: https://github.com/sissaschool/turborvb. In this work we have used Sandro Sorella's legacy version (v1.0.0), which can be found at this link: https://github.com/sissaschool/turborvb/releases/tag/v1.0.0.

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

## Acknowledgements

F.M., R.V., A.M.S. and M.C. acknowledge Dominik Marx for useful discussions. M.P., R.V., A.M.S. and M.C. thank the CNRS for the 80PRIME doctoral grant allocation. F.M. and M.C. acknowledge computational resources provided by the PRACE project number 2016163936. M.C. thanks GENCI for providing computational resources at TGCC and IDRIS under the grant number 0906493, and the Grands Challenges DARI for allowing calculations on the Joliot-Curie Rome HPC cluster under the project number gch0420. T.M. and M.C. are grateful to the European Centre of Excellence in Exascale Computing TREX-Targeting Real Chemical Accuracy at the Exascale, which partially supported this work. This project has received funding from the European Unions Horizon 2020 Research and Innovation programme under Grant Agreement No. 952165.

## Author contributions

R.V., A.M.S. and M.C. conceived the project. F.M., M.P., and M.C. carried out the PIMD simulations. F.M. and M.C. prepared the variational wave function used in the QMC calculations. All the authors analysed and interpreted the data. F.M., R.V., A.M.S. and M.C. wrote the manuscript, with the help from M.P. and T.M.

## Competing interests

The authors declare no competing interests.
