## [Peer Review File · Nature Communications]

REVIEWER COMMENTS

Reviewer #1 (Remarks to the Author):

This paper reports the results of quantum simulations of $\text{H}+(\text{H}_2\text{O})_6$ as a function of temperature. The calculations are of high quality, and the calculations provide new insights into how the cluster evolves with increasing temperature. The VMC method is employed to accurately describe the potential energy surface, and the nuclei are treated both classically and quantum mechanically. The paper is suitable for publication in Nature Communications. I identified only a couple of minor issues that the authors may want to address.

1. In the manuscript the authors state that the Eigen and Zundel forms of $\text{H}+(\text{H}_2\text{O})_6$. This is true only when vibrational ZPE is included. In the absence of ZPE, there are other low-lying isomers. The authors should make clear that their statement pertains to the case that ZPE is included.
2. The authors state that the hexamer is the smallest water cluster for which both Eigen and Zundel forms exist. This is not correct even $\text{H}+(\text{H}_2\text{O})_4$ exists in both Eigen and Zundel forms. See e.g., Phys. Chem. A 2017, 121, 12, 2386–2398
3. On line 192, the authors state that near $d = 2.38\text{\AA}$ the potential becomes quartic because the curvature changes sign. The curvature is given by the second derivative which is zero for a quartic potential. I don't believe it is appropriate to say the potential is quartic because the curvature changes sign.

Reviewer #2 (Remarks to the Author):

This paper describes a series of path integral simulations of the protonated water hexamer using an accurate quantum monte carlo potential to provide energies and forces. Additionally, a reduced dimensional model at zero temperature is used to study the proton transfer vibration and its coupling with the O-O stretching mode. The methods used are impressive and all simulation methodologies are state-of-the-art. The authors make several interesting observations on the basis of these simulations.

First, they point out that the classical and quantum simulations are rather different in their temperature dependence for the hexamer cluster. In the classical case, there is a consistent thermal expansion of

the O-O distance. In the quantum case, thermal expansion is very slow and the O-O distribution even seems to shorten at around 250K.

Additionally, the authors find that the proton transfer rate seems to be maximized at around 250-300K, which is counter to the expectation that proton transfer would increase with temperature. This effect arises due to the large changes in the underlying PES as temperature increases, as explained in the paper.

While the results presented in the paper are quite interesting for the hexamer, i.e. the observation of a “sweet-spot” for proton transfer rate around room temperature, its generality to proton transfer in ordinary systems is not obvious. The authors imply that some fundamental behavior is relevant to proton transfer in other contexts, but it could simply (and likely is) an artifact of a cluster model, in particular the peculiar geometry of the protonated hexamer and thus is not a general observation. For instance, it is known that the diffusion constant of protons in aqueous solution strictly increases with temperature. This contradicts the observations made in this paper which, if they applied to liquid water, would imply a maximum in the proton diffusion rate at some point. This is also easy to see based on the fact that when more waters are added to the protonated hexamer, the zundel structure is destroyed and the eigen geometry is recovered. Indeed, proton transfer is never barrierless except in the protonated water hexamer.

Specific Comments:

On lines 72-73, the short O-O regime is characterized as having a quadratic potential. This seems doubtful as the zundel structure of the protonated water dimer is extremely anharmonic.

On lines 190-191, the authors again describe the barrierless regime as quadratic. It is hard to judge whether or not this is true just from figure 4, which really doesn't look quadratic near the minimum. The authors should fit each of the potentials with a polynomial to get coefficients associated with each quadratic, cubic, and quartic pieces of the potential in each regime. This will provide a more quantitative measure of the anharmonicity of each potential and should be easy to do. While the ZPE-corrected potentials will be more harmonic, anharmonicity primarily appears in the kinetic energy (it is a 100% kinetic contribution in the morse potential for instance).

As a minor point, the reduced-dimensional potential which the authors construct is very similar to one recently used by Huchmala and McCoy and they should cite: *J. Phys. Chem. A* 2022, 126, 8, 1360–1368. Indeed that paper is quite relevant to the present work.

Reviewer #3 (Remarks to the Author):

In this work the authors perform simulations of protonated water hexamers using the QMC level of theory incorporating nuclear quantum effects. This work appears carefully and thoroughly carried out and I cannot identify any major methodological issues and some quite interesting conclusions are drawn. Particularly the low thermal expansion associated with NQEs and the suggestion of the importance of shortened Zundel in proton diffusion at room temperature.

I have some concerns that should be addressed prior to publication, however.

Firstly, the references to the literature seem quite out of date. For example, the only reference to MD simulations is to a Voth paper from 2008. A large amount of work has been done on this problem since then. See the 2021 JACS paper from Voth (10.1021/jacs.1c08552) and a 202 paper from Mundy (10.1021/acs.jpcc.0c02649) and the references within. I think there needs to be some effort to engage with this literature and place the findings of this work in the wider context. Also at 5 years old I don't think the 2D IR spectroscopy is "Very recent" anymore.

I believe there should also be some discussion of the implications of this for bulk phase, which is ultimately why this problem is important. Do the authors believe these conclusions can be transferred? How confident are they. In particular, I think it is worth discussing the view put forth by Voth that the "a distorted and dynamic Eigen cation as the most prevalent hydrated proton species" in solution. Is this inconsistent with their results. They don't need to agree but some engagement/discussion is necessary, I think.

The authors claim in the conclusion that these methods are "unprecedentedly accurate" It's not clear what evidence justifies this claim as far as I can see. Figure S1 shows the dimer binding energy where it is off by 20 %. This doesn't seem very impressive.

The authors point to a maximum in the proton diffusivity and provide an explanation of it. They imply that it may be of significance for biology. If they want to make this claim I would provide some evidence that this enhanced diffusivity is also observed experimentally in bulk water in this temperature range. Otherwise, it is not at all clear that it generalises.

The end with the statement: "thus call for further efforts to explore the temperature behaviour of the proton dynamics and transport" I think they should be more specific. What exactly do they want people to investigate. This is already a well studied field. What do they think is missing.

I couldn't find the length of time the simulations were run for in the main paper. I think it should be easy to find.

Referee #1

This paper reports the results of quantum simulations of $\text{H}^+(\text{H}_2\text{O})_6$ as a function of temperature. The calculations are of high quality, and the calculations provide new insights into how the cluster evolves with increasing temperature. The VMC method is employed to accurately describe the potential energy surface, and the nuclei are treated both classically and quantum mechanically. The paper is suitable for publication in Nature Communications. I identified only a couple of minor issues that the authors may want to address.

We thank the Referee for judging our work suitable for publication.

1. In the manuscript the authors state that the Eigen and Zundel forms of $\text{H}^+(\text{H}_2\text{O})_6$. This is true only when vibrational ZPE is included. In the absence of ZPE, there are other low-lying isomers. The authors should make clear that their statement pertains to the case that ZPE is included.

The Referee is right that there are other low-lying isomers. We believe we do not sample them because they are topologically and energetically harder to access from the Zundel-Eigen configurations from which our simulations are initialized. Indeed the barriers connecting these low-lying states can be large and require very long simulation time to explore them. We added a comment in the text about the existence of other low-lying isomers.

2. The authors state that the hexamer is the smallest water cluster for which both Eigen and Zundel forms exist. This is not correct even $\text{H}^+(\text{H}_2\text{O})_4$ exists in both Eigen and Zundel forms. See e.g., Phys. Chem. A 2017, 121, 12, 2386–2398

We thank the Referee for this correction. We changed the text in the two places where this sentence appears, by saying that the hexamer is one of the smallest water cluster where both Eigen and Zundel forms exist. We also cited the reference suggested by the Referee.

3. On line 192, the authors state that near $d = 2.38\text{\AA}$ the potential becomes quartic because the curvature changes sign. The curvature is given by the second derivative which is zero for a quartic potential. I don't believe it is appropriate to say the potential is quartic because the curvature changes sign.

The Referee is right. We changed the sentence, by specifying that the potential becomes quartic in a region where the curvature is zero before changing sign. Please, see also this comment in connection with the one of Referee #2.

Referee #2

This paper describes a series of path integral simulations of the protonated water hexamer using an accurate quantum monte carlo potential to provide energies and forces. Additionally, a reduced dimensional model at zero temperature is used to study the proton transfer vibration and its coupling with the O-O stretching mode. The methods used are impressive and all simulation methodologies are state-of-the-art. The authors make several interesting observations on the basis of these simulations.

We thank the Referee for her/his positive appreciation of our work and of its methodological aspects.

First, they point out that the classical and quantum simulations are rather different in their temperature dependence for the hexamer cluster. In the classical case, there is a consistent thermal expansion of the O-O distance. In the quantum case, thermal expansion is very slow and the O-O distribution even seems to shorten at around 250K.

Additionally, the authors find that the proton transfer rate seems to be maximized at around 250-300K, which is counter to the expectation that proton transfer would increase with temperature. This effect arises due to the large changes in the underlying PES as temperature increases, as explained in the paper.

While the results presented in the paper are quite interesting for the hexamer, i.e. the observation of a “sweet-spot” for proton transfer rate around room temperature, its generality to proton transfer in ordinary systems is not obvious. The authors imply that some fundamental behavior is relevant to proton transfer in other contexts, but it could simply (and likely is) an artifact of a cluster model, in particular the peculiar geometry of the protonated hexamer and thus is not a general observation. For instance, it is known that the diffusion constant of protons in aqueous solution strictly increases with temperature. This contradicts the observations made in this paper which, if they applied to liquid water, would imply a maximum in the proton diffusion rate at some point.

We thank the Referee for raising the question about the generality of the proton transfer description provided in our work. We agree with the Referee that the prediction of the “sweet-spot” found in the protonated water hexamer around 300 K should be quantitatively confirmed in larger clusters and in “ordinary systems”. This was the meaning of our conclusive sentence “This thus call for further efforts to explore the temperature behaviour of the proton dynamics and transport”, alluded here and quoted by Referee #3. However, we believe that our findings are not a simple artefact of the cluster model. Indeed, the interaction strength between the first solvation shell (the two water molecules closest to the solvated proton) and the second one (the other 4 molecules saturating the OH bonds of the inner shell) is remarkably close to the interaction taking place in bulk water at the same temperatures, as far as the pair correlation functions are concerned (see Fig. 1). It is clear that, while the inner water molecules are much closer to each other due to the enhanced interaction provided by the hydrated proton, the average distance of the molecules between the first and the second shell is much closer to the measured bulk water O-O distance. Thus, we believe that the embedding provided by the second shell is quite effective in mimicking the aqueous environment.

Motivated by the criticism raised by the Referee, we thoroughly investigated the experimental literature on bulk water, to look for an experimental confirmation of our transferability argument. We found the paper J. Phys. Chem. B **126**, 8791-8803 (2022), reporting the latest experimental results on the limiting conductivities assessment of the H_3O^+ and D_3O^+ species. Thanks to these measures, performed at 20 MPa, the excess molar conductivities due to proton hopping have been estimated. They show a non-monotonous behavior, similar to the one found in our calculations, with a peak located in the 420 K - 430 K range. In this temperature range and at the pressure conditions of the experiment, the water density is only 7-8% smaller than in standard conditions (source NIST). Thus, its regime is comparable to the one taking place in our cluster. We cited the experimental paper and added this information in the latest version of the manuscript. This is reassuring of the fact that our findings reveal features transferable to larger systems. It is also clear that for the transferability being fully quantitative, explicit calculations need to be done in the systems of interest. Larger systems are however out of reach at the moment, within the present framework and at this level of accuracy. See also the comment on the computational cost, detailed in one of the answers to Referee #3. In the latest version of the manuscript, we changed those parts addressing the bulk water limit, by placing our conclusions in a more critical perspective. This is also in relation with the comments of Referee #3.

Figure 1: Pair correlation functions involving one O atom in the protonated hexamer core (O_{core}) and another O atom belonging to the second solvation shell ($O_{\text{solvation}}$). The pair correlation functions are plotted for both $T=200$ K (left panel) and $T=300$ K (right panel). The long-dashed line is the location of the O-O distance in the hexamer cluster core, while the short-dashed line is the experimental O-O distance in bulk water reported in J. Chem. Phys. **151**, 044502 (2019). Its value at 200 K has been extrapolated from the experimental distance in bulk water at the lowest available temperatures (≈ 250 K). The solid line is the average peak position of the $O_{\text{core}}-O_{\text{solvation}}$ pair correlation function computed in our QMC-driven PIMD simulations.

Finally, we made the title more precise, to specify that we studied the properties of the protonated water hexamer, from which all our conclusions can be drawn.

This is also easy to see based on the fact that when more waters are added to the protonated hexamer, the zundel structure is destroyed and the eigen geometry is recovered. Indeed, proton transfer is never barrierless except in the protonated water hexamer.

We agree with the Referee that when more waters are added, the barrier increases. For the smallest clusters, the value of the barrier will increase with size. However, it will saturate when the size of the cluster will be larger than the polarization cloud of the hydrated proton, screened by the water environment. As already mentioned in our manuscript, in the Zundel complex the oxygen-oxygen distance is shorter than the one in the inner core of the protonated water hexamer. By consequence, the proton transfer is barrierless in the former case, while it has a barrier in the protonated hexamer cluster. Therefore, already in the protonated hexamer the proton transfer is not barrierless, although the zero point energy is larger than the barrier in its relaxed geometry. The height of the barrier is expected to increase further when other solvation shells are added. According to Agmon (J. Chem. Phys. **122**, 014506 (2005)), one more hydration layer is needed to reach the water bulk limit. From this viewpoint, the hexamer is not that far from this limit, and some relevant effects do already emerge at this size, as discussed also above, and reported in the Discussion section of the latest version of the manuscript.

Specific Comments: On lines 72-73, the short O-O regime is characterized as having a quadratic potential. This seems doubtful as the zundel structure of the protonated water dimer is extremely anharmonic.

We agree with the Referee about her/his comment on anharmonicity of the total potential. It is certainly anharmonic. However, in our previous comment, we were addressing the features of the potential *along the shuttling mode*. For shorter O-O distances, the potential along the shuttling mode becomes more and more harmonic. Please, see also the analysis below. We understand that our previous comment was unclear. We have thus changed the corresponding sentence accordingly, to make its meaning more transparent.

On lines 190-191, the authors again describe the barrierless regime as quadratic. It is hard to judge whether or not this is true just from figure 4, which really doesn't look quadratic near the minimum. The authors should fit each of the potentials with a polynomial to get coefficients associated with each quadratic, cubic, and quartic pieces of the potential in each regime. This will provide a more quantitative measure of the anharmonicity of each potential and should be easy to do. While the ZPE-corrected potentials will be more harmonic, anharmonicity primarily appears in the kinetic energy (it is a 100% kinetic contribution in the Morse potential for instance).

We thank the Referee for the suggestion of fitting the potentials with a polynomial to get coefficients associated to each contribution. This is done in the Supplementary Information (SI), Fig. S12, panels (a) and (b), for the quadratic and the quartic coefficients, respectively. The potential $v_{2D}(d_{O1O2}, \theta)$ becomes harmonic *around* its minimum as soon as the quadratic coefficient, $b_{Landau} = b_{Landau}(d_{O1O2})$, becomes positive. This happens for $d_{O1O2} \gtrsim 2.37 \text{ \AA}$ in the downfolded model based on QMC-driven MD data sampled at 200 K and 300 K. The classical 0 K minimum is located at $d_{symm} \approx 2.38 \text{ \AA}$, as mentioned in the manuscript. As soon as $b_{Landau}(d_{O1O2}) > 0$, the quadratic region along the shuttling V_{mode} can be defined as the one such that $\theta \lesssim b_{Landau}/c_{Landau}$. Thus, this region progressively extends as d_{O1O2} gets shorter. At $d_{O1O2} = 2.32 \text{ \AA}$, the shortest d_{O1O2} taken into account in this study, the quadratic region extends up to $\approx 0.13 \text{ \AA}$ around the minimum. The wave function there has certainly a larger spread, which means that some anharmonic effects are always present. However, for this short distance, the regime can be seen as *mainly* quadratic. In Figure 4, panel (a), we agree with the Referee that the overall potential there is not quadratic. This is expected because it is computed at $d_{O1O2} = 2.375 \text{ \AA}$, as reported in the panel, a distance that is exactly at the boundary between the strongly anharmonic "Elongated Zundel" regime and the "Short Zundel" one.

As a minor point, the reduced-dimensional potential which the authors construct is very similar to one recently used by Huchmala and McCoy and they should cite: J. Phys. Chem. A 2022, 126, 8, 1360–1368. Indeed that paper is quite relevant to the present work.

We thank the Referee for pointing this out. It is a very interesting work, indeed, relevant for the description of the two-dimensional (2D) potential proposed in the present manuscript. We cited now this paper in our work, when we talk about the adiabatic Born-Oppenheimer approximation that is used to solve the effective 2D potential.

Reviewer #3

In this work the authors perform simulations of protonated water hexamers using the QMC level of theory incorporating nuclear quantum effects. This work appears carefully and thoroughly carried out and I cannot identify any major methodological issues and some quite interesting conclusions are drawn. Particularly the low thermal expansion associated with NQEs and the suggestion of the importance of shortened Zundel in proton diffusion at room temperature.

We are glad that the Referee found our work interesting.

I have some concerns that should be addressed prior to publication, however.

Firstly, the references to the literature seem quite out of date. For example, the only reference to MD simulations is to a Voth paper from 2008. A large amount of work has been done on this problem since then. See the 2021 JACS paper from Voth (10.1021/jacs.1c08552) and a 202 paper from Mundy (10.1021/acs.jpcc.0c02649) and the references within. I think there needs to be some effort to engage with this literature and place the findings of this work in the wider context. Also at 5 years old I don't think the 2D IR spectroscopy is "Very recent" anymore.

We thank the Referee to suggest including the most up-to-date references on MD simulations. We added them in the Introduction and we thus put our work in a wider perspective, comprising these latest works. We also changed "very recent" in "recent", when talking about the 2D IR spectroscopy.

I believe there should also be some discussion of the implications of this for bulk phase, which is ultimately why this problem is important. Do the authors believe these conclusions can be transferred? How confident are they. In particular, I think it is worth discussing the view put forth by Voth that the “a distorted and dynamic Eigen cation as the most prevalent hydrated proton species” in solution. Is this inconsistent with their results. They don’t need to agree but some engagement/discussion is necessary, I think.

We agree with the Referee on her/his comment of the importance of the implications of our findings for larger systems, such as bulk phases. We invite the Referee to read the response provided to Referee #2, raising the same criticism. As for the presence of “a distorted and dynamic Eigen cation as the most prevalent hydrated proton species”, we carried out a detailed analysis of the populations of the three species, i.e. short Zundel, elongated Zundel and distorted Eigen, as defined in our paper. We report these details in a new section (Sec. S.VI) of the SI. Their temperature dependence confirms the findings already reported in the previous version of our manuscript, based on the 2D distribution functions. In the present version we then added a reference to the new material shown in Sec. S.VI. In our case, the most prevalent hydrated proton species at 300 K is the elongated Zundel. We believe that this strongly depends on the way these species are defined. In Voth’s paper, this is based on the number of unique special pairs, while in our case we defined them according to the O-O distance. Therefore, a direct one-to-one correspondence of the absolute populations is not straightforward. However, a comparison can be made on population *transfer* between the Zundel- and Eigen-like species. Indeed, Voth’s paper agrees with ours in an important aspect, namely the increase of the Zundel population in the hydrated proton system when one goes from classical to quantum nuclei at 300 K, as shown here in Fig. 2, and in the same figure (Fig. S16) reported in Sec. S.VI of SI. This is in accordance with

Figure 2: Population of the short Zundel, elongated Zundel and distorted Eigen species at 300 K, evaluated from both classical and quantum QMC-driven MD.

one of the NQEs explained in the paper, leading to the non-trivial temperature behaviour: the Eigen-like configurations are penalised by NQEs due to their larger zero-point energy, absent in classical simulations. We added this remark in the text, by citing explicitly Voth’s work.

The authors claim in the conclusion that these methods are “unprecedentedly accurate” It’s not clear what evidence justifies this claim as far as I can see. Figure S1 shows the dimer binding energy where it is off by 20 %. This doesn’t seem very impressive.

Reproducing the correct binding energy is always a hard task, because the electronic theory needs to interpolate between two very different regimes, namely the binding region, and the atomization region, considered as “strongly correlated”, where the atoms are pulled apart at very large distance. However, the system simulated here is never in the atomization limit for any of its possible fragments.

Quantum and thermal fluctuations taken into account in our MD/PIMD dynamics let the system evolve within, or in proximity of, the binding region (more precisely within 0.4 \AA from the equilibrium position for the O-O stretching mode), and never further away. In this region, the *variation* of the potential provided by our QMC energies is much more accurate than the binding energy, as it can be seen from Fig. S1. To further check the accuracy of our QMC variational ansatz in describing

Figure 3: O-O pair correlation function computed for the Zundel complex by means of different approaches: the QMC-driven PIMD using a wave function with a primitive and a O[6]H[2] GEO contracted Jastrow factor. The results are compared with a PIMD simulations based on CCSD(T) forces.

the O-H⁺-O bond, in Fig. 3 and in the new Section SII.1 of the SI we plot the O-O pair correlation function of the Zundel ion, computed at different levels of theory: CCSD(T) (X. Huang, B. J. Braams, J. M. Bowman, J. Chem. Phys. **122**, 044308 (2005)), a QMC-JAGP ansatz with a Jastrow expanded in a primitive basis (already used in JCTC **13**, 2400-2417 (2017)), and a QMC-JAGP ansatz with a Jastrow in a O[6]H[2] GEO contracted basis set (computed for the first time here and used to simulate also the protonated water hexamer). As one can see from Fig. 3, the three pair correlation functions are in a good statistical agreement between each other, showing a similar quality in describing the O-O potential energy surface around the minimum. We do agree however that a more toned-down expression for our "accuracy" is more suitable. Thus, we changed that accordingly in the manuscript, by specifying that it is comparable to the CCSD(T) level, and by referring the reader to the SI Sections I and II for benchmarks.

The authors point to a maximum in the proton diffusivity and provide an explanation of it. They imply that it may be of significance for biology. If they want to make this claim I would provide some evidence that this enhanced diffusivity is also observed experimentally in bulk water in this temperature range. Otherwise, it is not at all clear that it generalises.

As reported in one of our replies to Referee #2, enhanced proton diffusivity is indeed found experimentally in bulk water, but at a larger temperature range ($T \in [420 \text{ K}, 430 \text{ K}]$). Therefore, we changed some of our conclusions, in particular those about biology. Moreover, the last part of the abstract has also been changed accordingly, and the citation to the bulk water paper is added in the main manuscript. The relation between the cluster and the water bulk limit/aqueous systems is now discussed in several parts of the main text.

The end with the statement: "thus call for further efforts to explore the temperature behaviour of the proton dynamics and transport" I think they should be more specific. What exactly do they want people to investigate. This is already a well studied field. What do they think is missing.

As also discussed in the reply to Referee #2, in order to explore the temperature behaviour of the proton dynamics and transport in extended systems, one would need to go towards larger clusters or towards bulk water, by keeping the same accuracy as the one reached here in the water hexamer by means of QMC-driven PIMD. As we have shown, the level of accuracy delivered in the present work is indeed needed to fully resolve the properties of the solvated proton in the hexamer cluster. Thanks to our approach, this study reveals some new important features of the aqueous proton, such as the non-monotonous behavior of the proton diffusivity, also present in bulk water. However, to make our findings transferable to larger systems in a quantitative way, one would need to carry out explicit calculations at the same level of accuracy. At present, this becomes prohibitively expensive in larger systems, within this framework. However, the generation of machine learning (ML) potentials trained on QMC-quality or CC-quality sets bear promise of speeding up the calculations in larger clusters or in bulk water by orders of magnitude without sacrificing the accuracy. Despite being already an extremely well studied field, tackling it by means of this class of new ML potentials is still at a pioneering stage. We extended the ending comments of our paper by including this perspective.

I couldn't find the length of time the simulations were run for in the main paper. I think it should be easy to find.

We agree with the Referee that this is an important piece of information, which also reveals the challenge we faced by running these simulations. We now refer the reader to Tab. S4 of SI Section SII.2 for an exhaustive information of the computational cost for the various simulations carried out in this work. As one can see, the cost of the simulations usually exceeds 2M CPU hours for a given temperature and for the total number of iterations reached in our work.

REVIEWER COMMENTS

Reviewer #1 (Remarks to the Author):

I read the authors' responses to the reviewers and the revised MS. I believe that they did a good job at addressing the questions/issues raised by the reviewers.

I do believe that there is one additional thing that they might want to address. Most spectroscopic work on the protonated hexamer has been done at temperatures below 200K, and some of the experimental studies have been done at 40 K or below. I believe that on the time scale of the experiments there would be evaporation of monomers from the cluster at the higher temperatures considered in this paper. Presumably for the times the authors run their simulations they don't see evaporation. It would be useful to discuss the relevancy of the temperatures considered.

Reviewer #2 (Remarks to the Author):

The reported simulations by the authors are very technically impressive and undoubtedly required a lot of effort. However, the authors response have not been wholly satisfactory on the following points.

(1) the biggest issue is relevance. The gas phase water hexamer and its peculiar temperature dependence of the diffusivity is a real stretch for relevance to the condensed phase. In fact the observations of this paper (diffusion maximizes at a certain temperature), disagrees with the known temperature dependence of proton diffusion for liquid water which simply increases with temperature (never maximizes).

(2) The authors also claim that binding energies are difficult because of the need to consider atomization energies. One doesn't need to atomize the cluster to get a binding energy. Just break it into H₂O and H₃O⁺ fragments.

(3) Additionally, the agreement with CCSD(T) is not as good as claimed by the authors. The water dimer curve in the SI is off by ~0.5 to 1 kcal/mol which is what a good force field can do.

(4) The authors claim that the isomers are well localized, but a recent JCPLett 2023 does seem to refute this assertion.

Reviewer #3 (Remarks to the Author):

I am happy with the changes made to the manuscript by the authors and think the manuscript is improved.

I am still somewhat skeptical that this effect significant in the condensed phase but they have found some evidence that it may be which is interesting and significant enough in my opinion.

I also don't really agree with the claim comparable in accuracy to CCSD(T) it is still 20 % off and CCSD(T) is surely still much more accurate. I personally think it should be rephrased as sufficiently accurate to reproduce structural properties comparable to CCSD(T) or something similar. There are many applications where the 20% error in binding energy will be significant and it's important not to mislead readers.

Referee #1

I read the authors' responses to the reviewers and the revised MS. I believe that they did a good job at addressing the questions/issues raised by the reviewers.

We thank the Referee for positive assessment of our work.

I do believe that there is one additional thing that they might want to address. Most spectroscopic work on the protonated hexamer has been done at temperatures below 200K, and some of the experimental studies have been done at 40 K or below. I believe that on the time scale of the experiments there would be evaporation of monomers from the cluster at the higher temperatures considered in this paper. Presumably for the times the authors run their simulations they don't see evaporation. It would be useful to discuss the relevancy of the temperatures considered.

The Referee is right. We do not see any evaporation in the ≈ 30 ps time evolution of our simulations. We think that the most recent pump-probe techniques, which are able to resolve the response of the system on a shorter time frame than the simulated cluster time, will allow one to possibly detect the physics we are exploring even at temperatures around the proton diffusion maximum. However, in the latest version of our manuscript, we added a sentence to warn the reader about the difficulty of performing experiments on this system at the highest temperatures studied here.

Referee #2

The reported simulations by the authors are very technically impressive and undoubtedly required a lot of effort. However, the authors response have not been wholly satisfactory on the following points.

We thank the Referee for appreciating the technical aspects of our work, which allowed us to provide a fresh new viewpoint on the proton transfer problem, at least in the protonated water hexamer.

1. The biggest issue is relevance. The gas phase water hexamer and its peculiar temperature dependence of the diffusivity is a real stretch for relevance to the condensed phase. In fact the observations of this paper (diffusion maximizes at a certain temperature), disagrees with the known temperature dependence of proton diffusion for liquid water which simply increases with temperature (never maximizes).

In the previous version of the paper we have already clarified the elements based on which the hexamer can be considered a "precursor" (in terms of solvation properties of the charged proton) of the condensed phase. We have also brought about its limitations. This is why we highlighted the fact that our primary target has always been the protonated water hexamer and not the bulk water, technically unreachable within our framework by the present means. We believe

that our findings in the protonated water hexamer are relevant *per se*. Moreover, given the structure of proton solvation, they can be “stretched” to some extent to the condensed phase. This finds some echo in J. Phys. Chem. B **126**, 8791-8803 (2022), where the measured excess molar conductivities, which isolate the effect of proton hopping from the diffusion induced by the external AC electric field, show a non-monotonous behavior, similar to the one found in our calculations, with a peak located in the 420-430 K range (see Figure 1). However, we

Figure 1: Figure taken from J. Phys. Chem. B **126**, 8791-8803 (2022) (Fig. 8(a) in that paper), showing the excess limiting molar conductivity of the hydrated protium (H₃O⁺) and the hydrated deuterium (D₃O⁺).

understand that the relation with more extended aqueous systems can be a source of debate. For these reasons, we removed from the abstract claims about proton dynamics in the condensed phase inferred from our results. Moreover, in the Introduction we removed a sentence about the protonated water hexamer that made a too strong link with the hydrated proton in bulk water. Thus, in the latest version of the manuscript, uncritical claims about bulk water have been removed. Nevertheless, we believe that our findings in the protonated water hexamer will stimulate further research in these directions.

- The authors also claim that binding energies are difficult because of the need to consider atomization energies. One doesn't need to atomize the cluster to get a binding energy. Just break it into H₂O and H₃O⁺ fragments.

We are very sorry for this misunderstanding. We never atomized the cluster, indeed. In the Supplementary Information (SI), the dissociation energies we reported for the water dimer have been obtained by pulling apart the two water molecules and by relaxing their structure at frozen oxygen-oxygen distance. However, what we wanted to say in the previous response is that both the atomization limit and the H₂O + H₂O, or H₂O + H₃O⁺ dissociation limits bear the same difficulty, namely the “strong correlation regime”, where the charge is localized on the

charges. This is a significant part of our work due to quantum fluctuations of localized species. We speculate that the break

- Additionally, the agreement with CCSD(T) is not as good as claimed by the authors. The water dimer curve in the SI is off by 0.5 to 1 kcal/mol which is what a good force field can do. (H₂O)_nH⁺ required to form the Zundel cation (H₂O)_nH⁺

Despite the fact that the dissociation limit is off by 0.5-1 kcal/mol, the structural properties and quantum-thermal distributions of the nuclei in the range explored by thermal excitations around their equilibrium structures, relevant for the quantities computed in our work, are in a good agreement with CCSD(T)-based PES. We have already shown it for the zundel ion, from ab initio simulations at 25 °C, completed by comparing the oxygen-oxygen pair distribution functions, and by adding that analysis to the SI in our previous upgrade of the paper. In this second rebuttal, we have carried out a similar analysis for the water dimer. We have performed PMD simulations based on the newly generated CCSD(T)-precision q-AQUA potential (J. Phys. Chem. Lett. **13**, 5068-5074 (2022))

at a temperature of 200 K, and compared them with our QMC-driven PIMD calculations, based on the same wavefunction quality as the one used in the protonated water hexamer (Fig. 2). As

Figure 2: Comparison between PIMD simulations of the water dimer driven by the q-AQUA potential (CCSD(T)) and our variational wavefunction (QMC). Left panel: Radial distribution function $g(r)$ obtained at 200 K for the *intra*-molecular oxygen-hydrogen pairs summed up. Right panel: oxygen-oxygen $g(r)$ at the same temperature. In these simulations, we used 32 beads in both CCSD(T) and QMC cases. The time evolution has been taken as long as ≈ 70 ps, with a time step of 1 fs, in order to have a good statistics on the oxygen-oxygen $g(r)$, which is particularly noisy to compute in the water dimer.

one can see from Fig. 2, there is a nice agreement in the structural properties (peaks position) and the quantum-thermal distribution (peaks width) between CCSD(T) and QMC calculations at the explored temperature. We added this new figure in the latest version of the SI, and we further commented about the quality of our QMC ansatz in the main paper (see also comment of Referee #3).

- The authors claim that the isomers are well localized, but a recent JCPlett 2023 does seem to refute this assertion.

The Eigen structure described in JCPlett 2023 is different from our distorted Eigen, because the former structure (labeled “E2”) requires a complete reshuffling of one of the external water molecules from one side of the cluster to the other. Therefore, this process is kinematically much slower, and thus not-so-relevant for the Zundel distorted-Eigen reconversion rate that we report in our work. Indeed, the reshuffling of the external water molecule described in JCPlett 2023 goes through a quite flat and very wide region of the PES, as shown also by the presence of a transition state “TS2” slightly up in energy. We did not see this process in our calculations, which correspond to a time evolution of 50 ps at most. Instead, our distorted Eigen configurations, which are sampled due to quantum and thermal fluctuations, bear some resemblance with the “T2” zero-temperature structure, topologically much closer to the Zundel-like “Z1” configuration. However, a perfect correspondence between classical geometries and those stabilized by quantum-thermal effects is hard to make, owing to the relevance of nuclear quantum effects in this system, which lead to distorted structures by quantum anharmonicity. Nevertheless, we added this recent JCPlett reference to the latest version of our manuscript, when we write about the existence of other isomers in the protonated water hexamer, which however we do not sample, for kinematic/energetic reasons.

Reviewer #3

I am happy with the changes made to the manuscript by the authors and think the manuscript is improved.

We are glad that the Referee found the changes we made to the manuscript satisfactory.

I am still somewhat skeptical that this effect significant in the condensed phase but they have found some evidence that it may be which is interesting and significant enough in my opinion.

As mentioned in our reply to Reviewer #2, we removed from the abstract claims about proton dynamics in the condensed phase inferred from our results. Nevertheless, we share with Reviewer #3 his/her opinion on the significance of our results. Indeed, we believe that our findings in the protonated water hexamer will stimulate further research in these directions.

I also don't really agree with the claim comparable in accuracy to CCSD(T) it is still 20% off and CCSD(T) is surely still much more accurate. I personally think it should be rephrased as sufficiently accurate to reproduce structural properties comparable to CCSD(T) or something similar. There are many applications where the 20% error in binding energy will be significant and it's important not to mislead readers.

We agree with the Referee that the word "comparable" can be misleading. We have now rephrased the sentence, by highlighting that, despite our QMC wavefunction ansatz being $\approx 20\%$ off from CCSD(T) in the dissociation limit, it is sufficiently accurate to yield structural properties and quantum-thermal distributions around equilibrium in a good agreement with CCSD(T). In this regard, please, see also our reply to Reviewer #2, and the newly added plot in the SI for the water dimer at 200 K.

REVIEWERS' COMMENTS

Reviewer #2 (Remarks to the Author):

We thank the authors for being responsive to our criticisms, which have been fully addressed in the revised manuscript. No further changes are needed.

Reviewer #3 (Remarks to the Author):

I am satisfied with the changes made by the authors in response to my comments, and I believe the manuscript is suitable for publication.